# Two new polymorphic structures of human full-length alpha-synuclein fibrils solved by cryo-electron microscopy

Ricardo Guerrero-Ferreira[1†], Nicholas MI Taylor[2], Ana-Andreea Arteni[3,4], Pratibha Kumari[5], Daniel Mona[6], Philippe Ringler[1], Markus Britschgi[6], Matthias E Lauer[7], Ali Makky[8], Joeri Verasdonck[5], Roland Riek[5], Ronald Melki[4], Beat H Meier[5], Anja Böckmann[9*], Luc Bousset[4*], Henning Stahlberg[1*]

[1]Center for Cellular Imaging and NanoAnalytics (C-CINA), Biozentrum, University of Basel, Basel, Switzerland; [2]Structural Biology of Molecular Machines Group, Protein Structure & Function Programme, Novo Nordisk Foundation Center for Protein Research, Faculty of Health and Medical Sciences, University of Copenhagen, Copenhagen, Denmark; [3]Institut de Biologie Intégrative de la Cellule (I2BC), CEA, CNRS, Université Paris Sud, Université Paris-Saclay, Gif-sur-Yvette, France; [4]Institut Fancois Jacob (MIRCen), CEA and Laboratory of Neurodegenerative Diseases, CNRS, Fontenay-Aux-Roses, France; [5]Laboratory of Physical Chemistry, ETH Zurich, Zurich, Switzerland; [6]Roche Pharma Research and Early Development, Neuroscience and Rare Diseases Discovery and Translational Medicine Area, Neuroscience Discovery, Roche Innovation Center Basel, Basel, Switzerland; [7]Roche Pharma Research and Early Development, Therapeutic Modalities, Roche Innovation Center Basel, Basel, Switzerland; [8]Institut Galien Paris-Sud, CNRS, Université Paris-Sud, Université Paris-Saclay, Châtenay-Malabry, France; [9]Molecular Microbiology and Structural Biochemistry, Labex Ecofect, UMR 5086 CNRS, Université de Lyon, Lyon, France

*For correspondence:
anja.bockmann@ibcp.fr (AB);
Luc.BOUSSET@cnrs.fr (LB);
Henning.Stahlberg@unibas.ch (HS)

Present address: †Robert P Apkarian Integrated Electron Microscopy Core, Emory University School of Medicine, Atlanta, United States

**Abstract** Intracellular inclusions rich in alpha-synuclein are a hallmark of several neuropathological diseases including Parkinson's disease (PD). Previously, we reported the structure of alpha-synuclein fibrils (residues 1–121), composed of two protofibrils that are connected via a densely-packed interface formed by residues 50–57 (Guerrero-Ferreira, eLife 218;7: e36402). We here report two new polymorphic atomic structures of alpha-synuclein fibrils termed polymorphs 2a and 2b, at 3.0 Å and 3.4 Å resolution, respectively. These polymorphs show a radically different structure compared to previously reported polymorphs. The new structures have a 10 nm fibril diameter and are composed of two protofilaments which interact via intermolecular salt-bridges between amino acids K45, E57 (polymorph 2a) or E46 (polymorph 2b). The non-amyloid component (NAC) region of alpha-synuclein is fully buried by previously non-described interactions with the N-terminus. A hydrophobic cleft, the location of familial PD mutation sites, and the nature of the protofilament interface now invite to formulate hypotheses about fibril formation, growth and stability.

## Introduction

Lewy bodies (LB) and Lewy neurites (LN) are neuropathological hallmarks of Parkinson's disease and other Lewy body disorders. These intracellular neuronal features contain a cytoplasmic enrichment of the protein alpha-synuclein (α-Syn), thereby defining these diseases as synucleinopathies

(*Spillantini et al., 1998a*; *Spillantini et al., 1997*). Apart from this finding in the postmortem brain, the central role of α-Syn in Parkinson's disease (PD) is highlighted by the fact that certain mutations in the α-Syn gene (SNCA) cause familial forms of PD and other synucleinopathies (*Appel-Cresswell et al., 2013*; *Krüger et al., 1998*; *Lesage et al., 2013*; *Pasanen et al., 2014*; *Polymeropoulos et al., 1997*; *Zarranz et al., 2004*) and that duplication or triplication of the SNCA gene lead to either a sporadic or early-onset PD, respectively, in affected families (*Chartier-Harlin et al., 2004*; *Flagmeier et al., 2016*; *Fujioka et al., 2014*; *Ibáñez et al., 2004*; *Singleton et al., 2003*). The 14 kDa protein α-Syn is known to readily form fibrils in vitro (*Conway et al., 1998*; *Hashimoto et al., 1998*) and induce α-Syn inclusions when injected in model animals (*Mougenot et al., 2012*; *Peelaerts et al., 2015*).

Shahmoradian et al. have recently investigated the ultrastructure of LBs, using identification by fluorescence light microscopy and then correlative imaging by electron microscopy (CLEM). They found that a slight minority of analyzed LBs contained filamentous or dense proteinaceous structures, while the vast majority of LBs was primarily composed of membrane fragments (*Lewis et al., 2019*; *Shahmoradian et al., 2019*). However, even in those cases, LBs had been identified due to their enrichment of α-Syn (antibody LB509, recognizing residues 115–122 of α-Syn). *Moors et al. (2018)* recently studied the ultrastructure of LBs by super-resolution light microscopy, revealing an onion-like distribution of different forms of α-Syn in nigral LBs and LNs. Their work suggests LBs to be structured encapsulations of aggregated proteins and lipids. While the structure and building blocks of LBs and the putative involvement of aggregated forms of α-Syn in the formation of LBs or in toxicity towards neurons and glia appear to be more complex than previously thought, it is clear that the protein α-Syn plays a pivotal role in PD, and therefore in LB formation. The conformational state and impact of α-Syn protein may also differ strongly between synucleinopathies. *Peng et al. (2018)* reported α-Syn preparations from glial cytoplasmic inclusions from multiple system atrophy patients being much more potent in seeding α-Syn aggregation into cell cultures than α-Syn preparations than PD patients. A detailed understanding of the conformational space of structural polymorphs of α-Syn is important to move forward the discovery of diagnostic tools and therapeutics for synucleinopathies, including PD.

α-Syn protein consists of 140 amino acids. The N-terminus (residues 1–60) is rich in lysine residues and contains KTK lipid-binding motif repeats associated with vesicle binding (*George, 2002*; *George et al., 1995*; *Perrin et al., 2000*). It is also the region which contains all known SNCA familial PD mutations: A30P (*Krüger et al., 1998*), E46K (*Zarranz et al., 2004*), H50Q (*Appel-Cresswell et al., 2013*; *Proukakis et al., 2013*), G51D (*Lesage et al., 2013*), A53E (*Pasanen et al., 2014*), and A53T (*Polymeropoulos et al., 1997*). The central region (residues 61–95) is the non-amyloid-ß component (NAC region) (*Giasson et al., 2001*; *Uéda et al., 1993*), which is essential for α-Syn aggregation (*Li et al., 2002*). The related protein ß-synuclein (ß-syn) (*Jakes et al., 1994*; *Stefanis, 2012*) lacks a stretch of 12 aminoacid residues within the NAC region (residues 71–82) and is unable to form fibrils.

The highly unstructured C-terminus of α-Syn (residues 96–140) can bind calcium and it is populated by negatively charged residues (*Li et al., 2007*; *Post et al., 2018*; *Vilar et al., 2008*). Truncation of this domain may play a role in α-Syn pathology by promoting fibril formation (*Crowther et al., 1998*; *Li et al., 2005*; *Liu et al., 2005*; *Wang et al., 2016*) and being involved in Lewy body formation (*Dufty et al., 2007*; *Mahul-Mellier et al., 2019*; *Prasad et al., 2012*). Inhibition of C-terminal truncation has also been shown to reduce neurodegeneration in a transgenic mouse model of Multiple System Atrophy (MSA) (*Bassil et al., 2016*).

Amyloid fibrils, even within a single sample, may exhibit heterogeneous conformations, referred to as polymorphism. Different polymorphs can be distinguished based on their diameter, their twist, how many protofilaments form a fibril (*Riek, 2017*), their behavior under limited proteolysis, their appearance under fiber diffraction (*Bousset et al., 2013*), or their structure using NMR or cryo- EM (*Close et al., 2018*; *Fändrich et al., 2018*; *Meier and Böckmann, 2015*). The polymorphs can be structurally different at the level of the protofibrils, or in the way the protofilaments assemble. Specifically for α-Syn fibrils, structural heterogeneity has been observed by solid-state NMR (*Bousset et al., 2013*; *Comellas et al., 2012*; *Comellas et al., 2011*; *Gath et al., 2012*; *Gath et al.,*

*2014a*; *Gath et al., 2014b*; *Heise et al., 2005*; *Lv et al., 2012*; *Verasdonck et al., 2016*; *Vilar et al., 2008*), quenched hydrogen/deuterium (H/D) exchange data (*Vilar et al., 2008*) and cryo-EM (*Guerrero-Ferreira et al., 2018*; *Li et al., 2018a*; *Li et al., 2018b*). A detailed understanding of the conformational space that structural polymorphs of α-Syn do access is central to not only understand how the same polypeptide chain can fold into different structures, but also to thoroughly characterize materials used for in vitro and in vivo experiments.

Our recent work on the structure of recombinant α-Syn(1-121), using cryo-electron microscopy (cryo-EM) (*Guerrero-Ferreira et al., 2018*) and other investigations on full-length α-Syn (*Li et al., 2018a*; *Li et al., 2018b*) revealed α-Syn fibrils in a structure composed of two protofilaments that buried the sites associated with familial PD in the interface region between the two protofilaments. This is here termed α-Syn polymorph 1a. Li et al. (*Li et al., 2018a*) reported an additional polymorph, here termed α-Syn polymorph 1b, in which the interface between virtually identical protofilaments is different. Previous reports on the structure of α-Syn fibrils by micro-electron diffraction (microED; *Rodriguez et al., 2015*) or solid-state NMR (*Tuttle et al., 2016*) either focused on small peptides, or did not describe the two protofilaments and the interface region.

Currently, no high-resolution structures of patient derived α-Syn have been obtained. Previous studies looking at the overall morphology of these types of samples, offered a first glance to fibril variability within a sample and among synucleinopathies. Filaments immunolabeled by α-Syn antibodies have been described as straight, unbranched fibrils, with widths of 5 nm or 10 nm and with varying patterns of conjugated gold depending on the specific antibody used (*Crowther et al., 2000*). Filaments extracted from cingulate cortex of patients with Dementia with Lewy Bodies (DLB) were also positive for α-Syn immunolabeling and revealed comparable morphologies (*Spillantini et al., 1998a*). In contrast, filaments extracted from Multiple System Atrophy (MSA) brains exhibited larger diameter spanning from 5 nm to 18 nm with two morphologies described by Spillantini et al. (*Spillantini et al., 1998b*) as 'straight'' and 'twisted'.

Here, we report new cryo-EM polymorphic structures of in vitro generated amyloid fibrils of α-Syn (denoted polymorphs 2a and 2b). Our structures reveal remarkable differences to the previously solved α-Syn polymorphs, and inform new hypotheses to explain the mechanism of and factors involved in in vitro amyloid fibril assembly, the potential role of familial PD mutations on fibril structure, and contribute to our understanding of amyloid fibril polymorphism.

## Results and discussion

### The structure of α-Syn fibril polymorphs

Fibrils of recombinant, full-length, human α-Syn were prepared using the conditions summarized in *Table 1*. Preformed fibrils (PFFs) were quick-frozen in holey carbon-coated copper grids and imaged with a Titan Krios electron microscope at 300kV, equipped with a Quantum-LS energy filter. Micrographs were acquired with a K2 Summit direct electron detector, drift-corrected and dose-weighted through the FOCUS interface (*Biyani et al., 2017*).

Helical image processing of 100'323 fibril segments extracted from 1'143 micrographs, revealed the presence of two distinct fibril polymorphs at the step of 3D classification. These polymorphs, termed α-Syn polymorph 2a and α-Syn polymorph 2b to distinguish them from the previously described α-Syn fibrils (α-Syn polymorphs 1a and 1b; *Guerrero-Ferreira et al., 2018*; *Li et al., 2018a*; *Li et al., 2018b*), were separately refined, resulting in 3D reconstructions at overall resolutions of 3.0 Å and 3.4 Å respectively (*Table 2*, *Table 3*, *Figure 1*, *Figure 1—figure supplement 1* and *Video 1*). The left-twisting handedness of the fibrils was confirmed by AFM imaging, as done previously (*Guerrero-Ferreira et al., 2018*). The maps show clear side-chain densities and ß-strand separation along the helical axis, and indicate that both fibril types are formed by two protofilaments of approximately 5 nm diameter, which are composed of distinct rungs of density.

Refined atomic models of the fibril cores indicate that polymorph 2a and polymorph 2b in terms of their local atomic structure share a common protofilament kernel, and which is clearly distinct from the one described previously in polymorphs 1a and 1b. However, the packing between the two protofilaments is different. Therefore, α-Syn fibrils exhibit assembly polymorphism as defined by *Riek (2017)* and described in Tau between paired helical filaments (PHFs) and straight filaments (SFs) (*Fitzpatrick et al., 2017*). In each protofilament of the new α-Syn polymorph 2a, successive

**Table 1.** Growth conditions for α-Syn fibrils.

| Study | Buffer composition | pH | Temp–erature | Time | Concen–tration | Method | α-Syn type | Poly–morph | PDB |
|---|---|---|---|---|---|---|---|---|---|
| This study | 50 mM Tris-HCl 150 mM KCl | 7.5 | 37°C | 1 week (600 r.p.m.) | 700 μM | Cryo-EM + NMR | Full-length ('named fibrils') | 2a, 2b | 6ssx 6sst |
| (*Guerrero-Ferreira et al., 2018*) | DPBS (Gibco) 2.66 mM KCl, 1.47 mM KH2PO4, 137.93 mM NaCl, 8.06 mM Na2HPO4 | 7 to 7.3 | 37°C | 5 days (1000 r.p.m.) | 360 μM (5 mg/mL) | Cryo-EM | Truncated (1-121) | 1a | 6h6b |
| (*Li et al., 2018a*) | 50 mM Tris, 150 mM KCl, 0.05% NaN3 | 7.5 | 37°C | 3 days (900 r.p.m.) | 500 μM | Cryo-EM | Full-length, N-terminal acetylated | 1a | 6a6b |
| (*Li et al., 2018b*) | 15 mM tetrabutyl–phosphonium bromide | Not speci–fied | Room tempe–rature | 14–30 days (quiescent) | 300 μM | Cryo-EM | Full-length | 1a,b | 6cu7 6cu8 |
| (*Tuttle et al., 2016*) | 50 mM sodium phosphate 0.12 mM EDTA 0.02% sodium azide (w/v) | 7.4 | 37°C | 3 weeks (200 r.p.m.) | 1000 μM (15 mg/mL) | NMR | Full-length | 1a | 2n0a |
| (*Rodriguez et al., 2015*) | 5 mM lithium hydroxide 20 mM sodium phosphate 0.1 M NaCl | 7.5 | 37°C | 72 hr | 500 μM | Micro-ED | Peptides: SubNACore, NACore, PreNAC | | 4rik 4ril 4znn |
| (*Rodriguez et al., 2015*) | 50 mM Tris 150 mM KCl | 7.5 | 37°C | 72 hr | 500 μM | No structure | Full-length | | |
| (*Gath et al., 2014a*) | 50 mM Tris-HCl 150 mM KCl | 7.5 | 37°C | 4 days (600 r.p.m.) | 300 μM | NMR secondary structure | Full-length | 2 | |
| (*Gath et al., 2012*) | 5 mM Tris-HCl | 7.5 | 37°C | 7 days (600 r.p.m.) | 300 μM | NMR secondary structure | Full-length | Different from 1, 2 | |
| (*Verasdonck et al., 2016*) | 5 mM NaPO4 | 9 | 37°C | 4 days (600 r.p.m.) | 300 μM | NMR secondary structure | Full-length | 1 | |
| This study | DPBS (Gibco): 2.66 mM KCl, 1.47 mM KH2PO4, 137.93 mM NaCl, 8.06 mM Na2HPO4 | 7 to 7.3 | 37°C | 5 days (1000 r.p.m.) | 360 μM (5 mg/mL) | Cryo-EM | Full-length, E46K | 2a | |
| This study | DPBS (Gibco): 2.66 mM KCl, 1.47 mM KH2PO4, 137.93 mM NaCl, 8.06 mM Na2HPO4 | 7 to 7.3 | 37°C | 5 days (1000 r.p.m.) | 360 μM (5 mg/mL) | Cryo-EM | Full-length, N-terminal acetylated | 2a | |
| This study | DPBS (Gibco): 2.66 mM KCl, 1.47 mM KH2PO4, 137.93 mM NaCl, 8.06 mM Na2HPO4 | 7 to 7.3 | 37°C | 5 days (1000 r.p.m.) | 360 μM (5 mg/mL) | Cryo-EM | Full-length, Phosphorylation at position S129 | 2a | |

rungs of ß-strands are related by helical symmetry with a rise of 4.8 Å and a twist of −0.80° with the subunits within the two protofilaments packed in the same plane, facing each other (*Figure 1D*, *Figure 1—figure supplement 2C*) in two-fold symmetry. Distinctively, the two protofilaments in α-Syn polymorph 2b are offset by 2.4 Å in height between each other, related by an approximate $2_1$ screw symmetry with a twist of 179.55° (*Figure 1E*, *Figure 1—figure supplement 2D*). In α-Syn polymorph 2a, residue K45 of one protofilament forms a salt-bridge with residue E57 of the other protofilament, and vice-versa (*Figure 1B*, *Figure 1—figure supplement 2A*). In α-Syn polymorph 2b, the interaction between protofilaments occurs only through salt-bridges between residues K45 and E46 from adjacent protofilaments (*Figure 1C*, *Figure 1—figure supplement 2B*).

**Table 2.** Cryo-EM structure determination statistics.

| | E46K mutant | Phosphorylated | N-terminal acetylated | α-Syn polymorph 2a | α-Syn polymorph 2b |
|---|---|---|---|---|---|
| **Data Collection** | | | | | |
| Pixel size [Å] | 0.831 | 0.831 | 0.831 | 0.629 | 0.629 |
| Defocus Range [μm] | −0.8 to −2.5 | −0.8 to −2.5 | −0.8 to −2.5 | −0.8 to −2.5 | −0.8 to −2.5 |
| Voltage [kV] | 300 | 300 | 300 | 300 | 300 |
| Exposure time [s per frame] | 0.2 | 0.2 | 0.2 | 0.2 | 0.2 |
| Number of frames | 50 | 50 | 50 | 50 | 50 |
| Total dose [e⁻/Å²] $[e^-/\text{Å}^2]$ | 69 | 69 | 69 | 69 | 69 |
| **Reconstruction** | | | | | |
| Box width [pixels] | 280 | 280 | 280 | 280 | 280 |
| Inter-box distance [pixels] | 28 | 28 | 28 | 28 | 28 |
| Micrographs | 843 | 1'887 | 948 | 1'143 | 1'143 |
| Manually picked fibrils | 2'702 | 5'095 | 3'751 | 5'233 | 5'233 |
| Initially extracted segments | 65'893 | 107'144 | 43'276 | 100'323 | 100'323 |
| Segments after 2D classification | 50'514 | 107'126 | 35342 | 100'193 | 100'193 |
| Segments after 3D classification | 50'514 | 21'685 | 35342 | 19'937 | 3'989 |
| Resolution after 3D refinement [Å] | 4.65 | 4.65 | 4.75 | 3.34 | 3.75 |
| Final resolution [Å] | 4.56 | 4.31 | 4.39 | 2.99 | 3.39 |
| Estimated map sharpening B-factor [Å²] | −208.6 | −104.3 | −178.1 | −67.1 | −76.4 |
| Helical rise [Å] | 4.85 | 4.84 | 4.78 | 4.80 | 2.40 |
| Helical twist [°] | −0.79 | −0.77 | −0.71 | −0.80 | 179.55 |

**Table 3.** Model building statistics.

| | α-Syn polymorph 2a | α-Syn polymorph 2b |
|---|---|---|
| Initial model used [PDB code] | 6ssx | 6sst |
| Model resolution [Å] (FSC = 0.143) | 2.98 | 3.4 |
| Model resolution range [Å] | 2.98 | 3.1 |
| Map sharpening B-factor [Å²] | −67.1 | −76.4 |
| Model composition<br>Non-hydrogen atoms<br>Protein residues<br>Ligands | 4900<br>730<br>0 | 4900<br>730<br>0 |
| B-factors [Å²] (non-hydrogen atoms)<br>Protein<br>Ligand | 48.18<br>-<br> | 70.32<br>-<br> |
| R.m.s. deviations<br>Bond lengths [Å]<br>Bond angles [°] | 0.008<br>0.938 | 0.009<br>0.941 |
| Validation<br>MolProbity score<br>Clashscore<br>Poor rotamers [%] | 2.43<br>6.14<br>4.26 | 2.11<br>5.73<br>2.13 |
| Ramachandran plot<br>Favored [%]<br>Allowed [%]<br>Disallowed [%] | 87.39<br>12.61<br>0.00 | 89.86<br>10.14<br>0.00 |

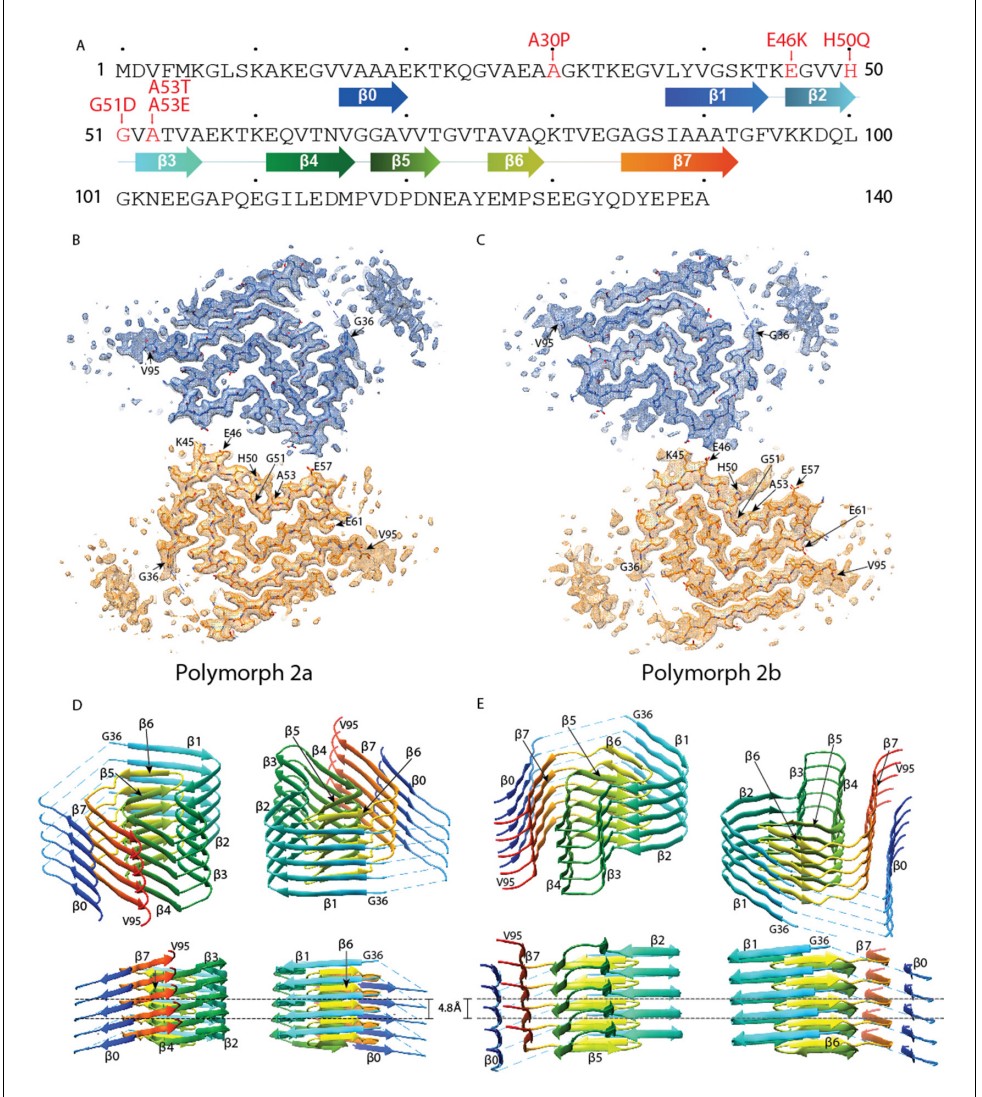

**Figure 1.** Cross-sections of the α-Syn polymorph 2a and 2b cryo-EM structures. (**A**) Sequence of human α-Syn with familial PD mutation sites indicated in red. ß strands are indicated by arrows colored from blue to orange. Cryo-EM densities and atomic models of polymorph 2a (**B**) and polymorph 2b (**C**) of α-Syn. Each cryo-EM map shows two protofilaments (blue and orange) forming a fibril. PD-associated mutations sites, and first and last residues of the NAC regions are indicated. (**D** and **E**) Rainbow rendering views of the secondary structure elements in five successive rungs of both polymorphs. A view perpendicular to the helical axis is shown to illustrate the height differences in a single α-Syn fibril. Colors correspond to the arrows in the sequence displayed in panel (**A**). The online version of this article includes the following figure supplement(s) for figure 1:

**Figure supplement 1.** Local resolution estimation and FSC curves.

**Figure supplement 2.** Interface regions between two protofilaments of the α-Syn polymorph 2a and 2b.

**Figure supplement 3.** NMR identification of the residues forming the N-terminal beta strand.

For polymorph 2a (PDB ID 6SSX) and polymorph 2b (PDB ID 6SST), each α-Syn molecule within a protofilament is composed of eight in-register parallel ß-strands (ß0–7; *Figure 1A,D and E*): residues 16–20 (ß0), 38–44 (ß1), 46–50 (ß2), 52–56 (ß3), 61–66 (ß4), 68–72 (ß5), 76–79 (ß6), and 85–92 (ß7). These are separated by either a lysine (L45) between ß1 and ß2, glycine residues (that is, G51 between ß2 and ß3, and G67 between ß4 and ß5) or arches (that is, E57-K60 between ß3 and ß4, G73-T75 between ß5 and ß6, and K80-G84 between ß6 and ß7). The NAC region encompasses ß-strands ß4 to ß7 and it appears entirely surrounded by densities in our cryo-EM map with ß1 to ß3 on one side of the fibril and additional densities on the other side. Minor differences between

**Alpha-synuclein Polymorphs**

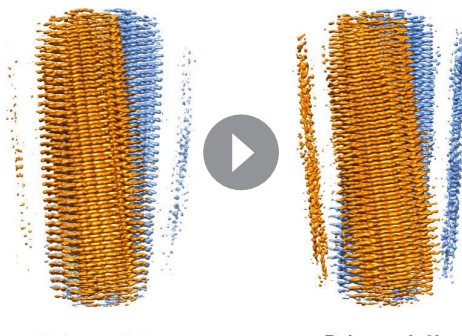

Polymorph 2a          Polymorph 2b

**Video 1.** Comparison of cryo-EM maps of α-Syn fibril polymorphs. Cryo-EM reconstructions of α-Syn fibrils at 3.0 Å (polymorph 2a) and 3.4 Å (polymorph 2b) resolution detailing the interaction between two protofilaments (blue and orange) in each fibril, the 4.8 Å spacing between ß-strands and the topology of α-Syn monomers within a single protofilament.
https://elifesciences.org/articles/48907#video1

polymorphs 2a and 2b are observed in the turn between ß3 and ß4 (*Figure 1D and E*, *Figure 2A and C*).

As described for α-Syn polymorph 1a (*Guerrero-Ferreira et al., 2018*), and also shown in patient-derived Tau filaments (*Fitzpatrick et al., 2017*), there are considerable changes in the height of the α-Syn monomer along the helical axis, with the highest point being ß3 and the lowest ß4 (*Figure 1D and E*).

α-Syn fibrils in polymorph 1a are formed by ß-strands that are arranged in bent β-arches, running along the length of each protofilament, previously described as a Greek key-like architecture (*Guerrero-Ferreira et al., 2018*; *Li et al., 2018a*; *Li et al., 2018b*; *Tuttle et al., 2016*). Also, the polymorphs 2a and 2b reported here present bent β-arch motifs. However, the components and the orientation of ß-strands contributing to the motifs are now radically different. In polymorphs 2a and 2b, two bent β-arches orient back-to-back, one showing a single, and the other showing two bends. The first β-arch is formed by strands 1/6, 2/5 and 3/4, with bends located at residues K45/V74, G51/G67, and the tip formed by K58. The strands interact through mainly hydrophobic, but also polar clusters. The second bent β-arch shares β-strands 4, 5 and 6 with the first one, and comprises in addition β-strand 7; again, interactions within the arch are of hydrophobic and polar type. This arch shows only one bend, at residues G67/G84.

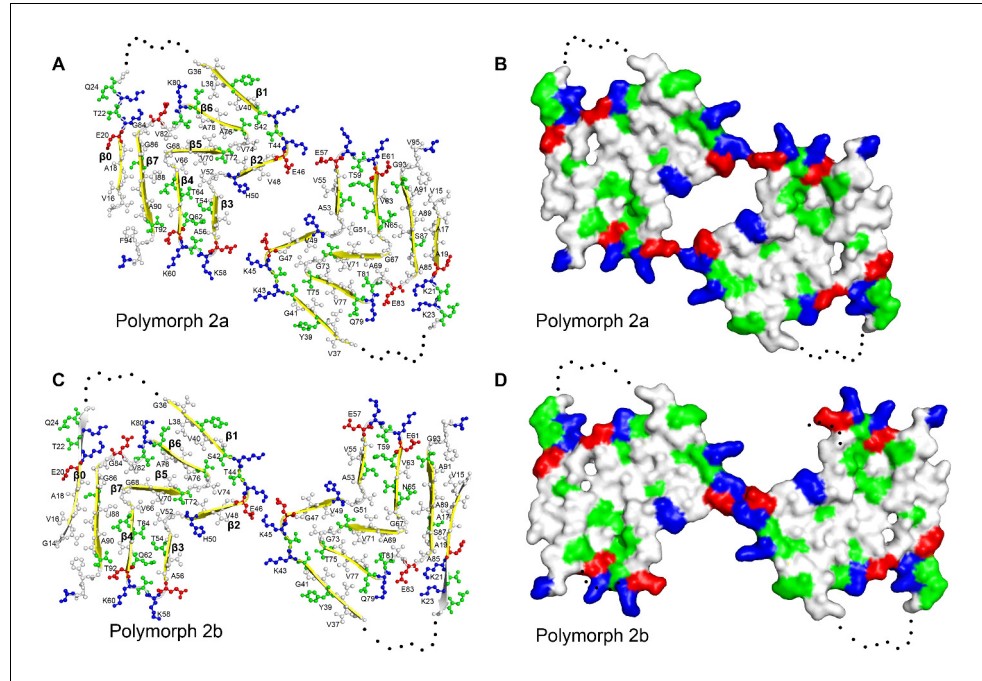

**Figure 2.** Structure and distribution of amino acids in the new α-Syn fibril polymorphs. Amino acids are colored in blue for positively charged, in red for negatively charged, in green for polar (including glycine), and in white for hydrophobic residues. Even and odd numberings are given on one monomer each. (A and C) Backbone structure. (B) and (D) Surface view.

The structure shares the extended use of glycine residues to form turns, and the hydrophobic and polar clusters forming the fibril core with other fibril structures (as for instance amyloid-β; *Gremer et al., 2017*; *Schütz et al., 2015*; *Wälti et al., 2016*), which contribute to protofilament stability, as also previously shown for α-Syn polymorph 1a (*Guerrero-Ferreira et al., 2018*; *Li et al., 2018a*; *Li et al., 2018b*).

Similar to α-Syn polymorph 1a, hydrophilic clusters are found at the periphery of the fibril (*Figure 2*). However, in contrast to α-Syn polymorph 1a, where the protofilament interface is formed by a hydrophobic steric-zipper geometry, in polymorphs 2a and 2b the interface is formed by salt bridges.

Interestingly, residue I88 marks the beginning of a hydrophobic area composed of residues A89, A90, A91, F94 and V95, which contribute to the stabilization of an additional beta-strand density that is clearly visible in the cryo-EM maps of both polymorphs (*Figure 1B and C*). We propose that this interacting region corresponds in polymorph 2a to a hydrophobic stretch formed by residues V16 to E20, and here use for it the term β0. This region was previously shown to be an isolated ß-strand identified in the N-terminus of α-Syn fibrils by solid-state NMR on equivalent fibril preparations (*Bousset et al., 2013*; *Gath et al., 2014a*) (*Figure 1—figure supplement 3A*). The localization of the stretch can be derived from cross peaks present in NMR 2D PAR (*De Paëpe et al., 2008*) spectra that connect S87 to A17, A18 and A19 (*Figure 1—figure supplement 3B*). Cross peaks in PAR spectra are indicative of proximities smaller than 6–7 Å between the spins, as illustrated in *Figure 1—figure supplement 3D*, which shows other meaningful structural restraints identified. This localizes A17, A18 and A19 in proximity to S87 (*Figure 1—figure supplement 3C*). The orientation of the N-terminal β0-strand is then given by both the absence in the spectra of cross peaks between S87 and V16, as well as clear side chain density fitting K21 in the EM map. Interestingly, no signals were observed in the NMR spectra for residues M1-V15, K21-V37 and V95-A140 in the α-Syn fibril structure. These regions correspond precisely to the stretches in our cryo-EM structure for which model building is not possible due to lower resolution, suggesting that these regions are indeed disordered in this polymorph. Also, no NMR peak doubling, although present for a subset of resonances (*Gath et al., 2014a*), was observed for residues at the filament interface, and no distinction could be made between polymorphs 2a and 2b, indicating that the structures of the monomers in the polymorphs 2a and 2b are very close to each other. The interruption of the polypeptide chain between strands ß0 and ß1 in polymorph 2a and 2b due to lack of clear density is compatible with a connection to the following strands of the same layer or alternatively to the neighboring layer.

For both α-Syn polymorph 2a and 2b, differences in height within residues of a protofilament reveal a hydrophobic area (*Figure 2*) akin to the hydrophobic cleft described for α-Syn polymorph 1a (*Guerrero-Ferreira et al., 2018*). It is composed of residues Q62 to V74 (ß4-ß5) (*Figure 1D and E*), and located between ß-strands ß2/ß3 and ß6/ß7. These residues correspond to a stretch of exclusively hydrophobic or polar residues, completely devoid of charged amino acids. Consistent with these results and with the concept that the hydrophobic core is essential for assembly, is the finding by *El-Agnaf et al. (1998)* that, within the NAC region, residues E61-A78 are the amyloidogenic component. The hydrophobic cleft in α-Syn polymorphs 2a and 2b contrasts with the one found in α-Syn polymorph 1a, where intermolecular interactions involving residues V74-V82 may be the initial binding event responsible for fibril elongation (*Guerrero-Ferreira et al., 2018*).

Our structural cryo-EM analysis of α-Syn fibrils prepared with E46K mutant or phosphorylated and N-terminally acetylated protein shows these also to be composed of two protofilaments and having an overall diameter of 10 nm (*Figure 3*). The lower order of these fibrils prevented a separation of the α-Syn rungs along the fibril axis, but still allowed discerning the separation of individual ß-sheets. As with polymorphs 1a, 2a and 2b, ß-sheets in α-Syn fibrils from E46K mutant, phosphorylated or N-terminally acetylated protein are arranged forming the characteristic bent β-arch like shape, and an arrangement closely resembling that of polymorph 2a (*Figure 3*).

## Comparison with previous structures

At first glance, the back-to-back arranged β-arches of polymorphs 2a and 2b appear similar to that of polymorph 1a, wrongly proposing that a mere protofilament rearrangement defines the difference between them. However, closer inspection and comparison with secondary structure elements identified in previous NMR studies on equivalent fibril preparations (*Bousset et al., 2013*; *Gath et al., 2014a*) revealed that polymorphs 2a and 2b radically differ from the construction of

**Figure 3.** Cryo-EM cross-sections of fibrils, formed by E46K, p-S129 phosphorylated, and N-terminally acetylated α-Syn protein. Fibrils formed by E46K mutant α-Syn protein (**A**), Ser129 phosphorylated α-Syn protein (**B**), and N-terminally acetylated α-Syn protein (**C**) were analyzed by cryo-EM. Image processing did not allow reaching sufficient resolution for model building, but the cross-sections of the obtained 3D reconstructions are compatible with polymorph 2a for all three forms.

polymorphs 1. *Figure 4* shows the backbones of the polymorphs with different color codes for N-terminus, NAC region, and C-terminus, and *Figure 4—figure supplement 1* shows each 10 residues in a different color. When compared to polymorphs 1, the new structures of polymorphs 2a and 2b show an inverted bend of the first β-arch motif, comprising mainly the black and cyan segments in *Figure 4—figure supplement 1*. This motif is largely conserved between polymorphs 1a and 1b (*Li et al., 2018a*) (*Figure 4*, *Figure 4—figure supplement 1* and *Video 2*), where it forms part of the interfilament interface, once via the black segment in polymorph 1a, and once via the cyan/light green segments in polymorph 1b. In polymorphs 2, this β-arch is formed by an inverted amino-acid sequence: when the two motifs are superimposed, the chain runs from cyan to black in one case, in the other from black to cyan. This inversion profoundly changes the amino acid distribution within the arch between polymorphs 1 and 2. Also, the β-arch in polymorph two is extended with a second bend followed by a β-strand region, comprising part of the orange segment largely unobserved in polymorphs 1.

In polymorphs 1, the light green chain segment form, together with parts of the above-described first β-arch, a second bent β-arch, completed by the red segment. This motif again exists in polymorphs 2, but is located on the outside of the first arch, while in polymorph 1a it is located at the inside. Also, the arch is inverted, and the amino acid distribution again differs, as shown by the different arrangement of the colored segments. Polymorph 1b is devoid of this motif. A hydrophilic cavity identified in polymorph 1a (located in the bend between the black and light green segments) is also present in polymorphs 2, but there between the partly disordered pink/orange segments, and the red segment.

The different arrangements result in a radically changed interface between the protofilaments in polymorphs 1 and 2. Unlike the hydrophobic interaction between protofilaments in polymorph 1a by the black segment, and in 1b by the cyan and light green segments, the interface in the new polymorphs 2a and 2b is formed by electrostatic interactions in the green segment, through salt-bridges between residues K45 and E57, or K45 and E46, respectively in 2a and 2b polymorphs (*Figure 1—figure supplement 2C and F*, *Figure 1—figure supplement 3*, *Figure 4—figure supplement 1*).

Furthermore, the presence of an additional ß-strand ß0 on the periphery of the fibril (in the blue segment) is a remarkable difference between polymorphs 1 and polymorphs 2. This stretch of residues, V16-E20, forms a hydrophobic steric-zipper geometry with residues S87-A91, which are part of ß7, at the end of the amyloidogenic NAC region. In polymorphs 2a and 2b, this results in the N-terminus wrapping around the fibril and enclosing the NAC region (cyan, light green and red segments). This buries the serine residue at position 87, a phosphorylation site located within the NAC region, inside this interface. This is in contrast to α-Syn polymorph 1a, where approximately 40 residues on both ends of the α-Syn fibril are flexible and surround the fibril with a fuzzy coat, leaving part of the NAC region (that is, K80 to V95, red and magenta segments), including S87, exposed (*Figure 1B and C*, *Figure 4—figure supplement 1B*).

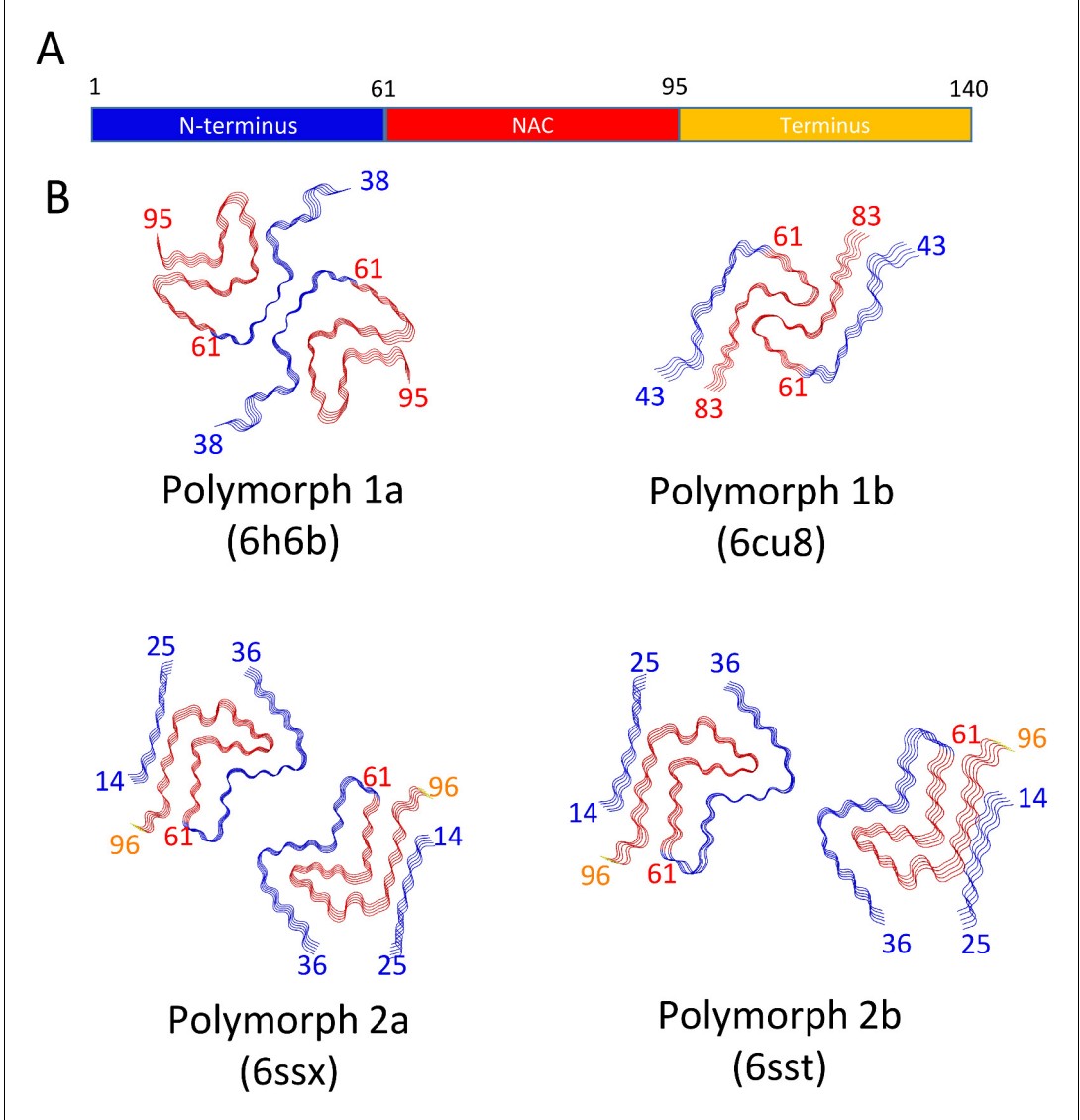

**Figure 4.** Schematic representation of α-Syn polymorphs. (**A**) Diagram representing α-Syn regions with the N-terminus in blue, the NAC region in red and the C-terminus in yellow. (**B**) Representation of α-Syn fibril polymorphs 1a (PDB ID 6h6b; *Guerrero-Ferreira et al., 2018*), 1b (PDB ID 6cu8; *Li et al., 2018a*), 2a (PDB ID 6rt0, this work), and 2b (PDB ID 6rtb, this work), highlighting the striking differences in protofilament folding in α-Syn polymorphs 1a and 1b, compared to α-Syn polymorphs 2a and 2b. The atomic models obtained by cryo-EM of α-Syn polymorph 1a, polymorph 1b (*Li et al., 2018a*; *Li et al., 2018b*) and α-Syn polymorphs 2a and 2b (this work). Protein Data Bank (PDB) accession numbers are indicated. The online version of this article includes the following figure supplement(s) for figure 4:

**Figure supplement 1.** Comparison between polymorphs 1 and 2.

## Origin of distinct α-Syn polymorphs

Presently, there are four different polymorphs of α-Syn fibrils known at atomic resolution, including the structures presented here. They can be classified into two groups with each group having a distinct fold (folds 1 and 2) (that is, at the protofilament level). Within each group there are two distinct protofilament packings or assembly polymorphisms (polymorphs a and b; *Figure 4*).

In an attempt to rationalize the origin of the distinct polymorphs, we first concentrate on the distinct folds at the protofilament level. All fold one structures (polymorphs 1a and 1b) were grown under buffer conditions comprising either a poly-anion (that is, phosphate with three negative charges and $N_3^-$ with having locally two negative charges) or a big chaotropic negative ion (that is

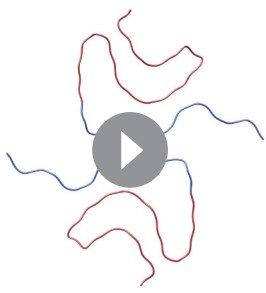

**Video 2.** Structural differences between α-Syn polymorphs. Atomic models of α-Syn fibrils represented as rounded ribbons with the N-terminus in blue and the NAC region in red.
https://elifesciences.org/articles/48907#video2

Br⁻), while polymorphs 2a and 2b of wildtype α-Syn were grown under phosphate-free conditions with the only anion being Cl⁻ (*Table 1*). Interestingly, there are adjacent to the salt-bridge H50-E57 three lysine residues in polymorph 1a at positions K43, K45 and K58 (the latter from the other protofilament) that are close in space. Between them, *Guerrero-Ferreira et al. (2018)* observed a density in the cryo-EM map, presumably a phosphate ion of poly-anionic nature that neutralizes the repulsion of the three positive charged residues (*Figure 4*). Albeit still of polymorph 1a type, in the presence of $N_3^-$ the structure is distinct in this area with K58 flipping inward into the cavity forming a salt-bridge with E61 attributed to the less poly-anion-like character of $N_3^-$ when compared with phosphate. In the presence of the chaotropic Br⁻, K58 is facing more the solvent with a closer distance between H50 and K45 (*Figure 4*). In the absence of a neutralizing poly-anion in this area of positively charged side chains or Br⁻ exerting its chaotropic property, it is unlikely that polymorph one can be obtained, indicating that in the presence of Cl⁻ as the only anion, a distinct polymorph must be formed.

The fibrils of the familial variant E46K (*Figure 3*) were also grown under phosphate conditions but adopt polymorph 2a fold. The polymorph 1a fold we obtained for the wildtype 1–121 form, introduces an electrostatic repulsion between K46 and K80, instead of the 46–80 salt-bridge stabilizing this polymorph. The structural features thus provide a rationale how changes corresponding to only a few kcal (or less) in the stability of polymorph 1a (such as absence/presence of phosphate, single point mutation), can lead to the protein adopting an entirely different polymorph. This finding also highlights the postulated flat energy landscape of fibril formation and the conformational promiscuity that comes along with it.

The fact that the polymorphs found both by Li et al. (*Li et al., 2018a*) and here by us are distinct, further accentuates this remark as they grew under similar conditions (except for the salt composition and additives, see *Table 1*). While the packing interfaces between polymorphs 1a and 1b are both hydrophobic, they still substantially differ, as in polymorph 1a residues V66-A78 form the interface, and in polymorph 1b residues H50-E57 form the interface (*Figure 4*). The difference is smaller in the case of polymorphs 2, where the protofilament interactions are of inter-protofilament salt-bridge character but are packed differently. It is evident that the energy differences between the two polymorphs are therefore very minute, but still the structures are distinct.

## Familial PD mutation sites in the new α-Syn polymorphs

A series of familial mutations have been identified in families with a history of PD. Based on *Fujioka et al. (2014)* and *Flagmeier et al. (2016)*, these mutations may lead to sporadic PD (A30P, E46K, A53E), early-onset (G51D, A53T) or late-onset (H50Q) forms of Parkinson's disease. The localization of the different sites in the structures is compared in *Figure 4—figure supplement 1*. Fundamentally different folds between polymorphs 1 and 2 place the familial PD mutation sites into an entirely different environment.

The E46K mutation has been found to promote α-Syn phosphorylation in mice (*Dettmer et al., 2017*), and in neuronal cells it showed to be toxic, with toxicity being enhanced by simultaneously mutating E35 and E61 to lysines (*Mbefo et al., 2015*). E46 holds a central role in polymorphs 1a and 2b. In polymorph 1a, where this motif resides at the beginning of one bent β-arch, E46 forms a stabilizing salt-bridge with K80. In contrast, in α-Syn polymorph 2b this residue is part of the protofilament interface, where the K45-E46 salt-bridge interface to E46-K45 from the other protofibril appears to be critical to the protofilament interface as these residues are the only interaction point between the protofilaments (*Figure 1—figure supplement 2*). Thus, protofilament interaction in this polymorph 2b manner would be unlikely since the mutation would induce a charge repulsion between lysines K45 and E46K from the two protofilaments. And indeed, the mutant E46K adopts

polymorph 2a, which result in two lysines (that is, K45 and K46) from one protofilament interacting with E57 in the other. Interestingly, the α-Syn E46K mutant fibril investigated here by cryo-EM, resulted in protofilament assembly corresponding to α-Syn polymorph 2a, confirming that the interaction between K45/K46 and E57 can indeed be maintained (*Figure 3*).

Heterozygous mutations in residues H50, G51, and A53 are associated with familial forms of PD (*Appel-Cresswell et al., 2013*; *Lesage et al., 2013*; *Pasanen et al., 2014*; *Polymeropoulos et al., 1997*). As we reported previously, in α-Syn polymorph 1a, these sites are an integral part of the interface region, contributing to the steric-zipper architecture and fibril stability, so that the mutations G51D, A53E and A53T are not compatible with polymorph 1a (*Guerrero-Ferreira et al., 2018*) (*Figure 4—figure supplement 1*). In contrast, in α-Syn polymorph 2a, these residues lie in the cavity formed between the two protofilaments, and in polymorph 2b they are surface-exposed where the two protofilaments interact (*Figure 4—figure supplement 1*). While these mutations are not in direct conflict with the formation of α-Syn fibril polymorphs 2a and 2b, the structures of fibrils formed with these mutations remain to be determined.

The A30P mutation leads to a form of PD with an age of onset between 54 and 76 years (*Flagmeier et al., 2016*; *Fujioka et al., 2014*; *Krüger et al., 1998*). In our structures of polymorphs 2a and 2b, a steric-zipper interface between V16-E20 and I88-A91 causes the α-Syn N-terminus to wrap around the NAC region. Under these conditions, residue A30 might be surface-exposed in a defined manner, as it forms part of the fibrillar core in the sense that it is on both sides linked to nearby structured regions. The disordered nature of the region between K21 and G37 results in weak density in the cryo-EM map, and makes accurate model building in this region difficult. Nevertheless, in α-Syn polymorph 1a, A30 is not part of the fibril core but instead is found in the disordered region corresponding to residues M1 to V37 (*Guerrero-Ferreira et al., 2018*).

## Post translational modifications (PTMs) and fibrillization in α-Syn polymorphs

Certain post-translational modifications of α-Syn associated with neuropathology inhibit the process of α-Syn fibril formation in vitro, suggesting that they are late events rather than occurring before protein aggregation (*Oueslati et al., 2010*). Most of these modifications take place at the C-terminal region (*Mahul-Mellier et al., 2019*), with the exception of acetylation and ubiquitination, which mostly affect residues in the N-terminal region. Ubiquitination alters mostly N-terminal lysines between residues 1 and 36, with K21, K23, K32 and K34 being the major sites for ubiquitin conjugation (*Nonaka et al., 2005*). The phosphorylated and acetylated forms determined here fold into polymorph 2a (*Figure 3*). In this polymorph, the amino-acid stretch, which appears disordered in polymorph 1a fibrils, is more distinct, as the interaction of ß-strand ß0 with the fibril core brings the N-terminal region back to the fibril. While the involvement of the N-terminal region has been suggested by NMR for two different polymorphs (*Bousset et al., 2013*; *Gath et al., 2012*; *Gath et al., 2014a*; *Gath et al., 2014b*), our structure for the first time shows how this N-terminal region that is important in the context of post-translational modifications, can be positioned in the protofilament.

## Implications for fibril preparation protocols

When pre-formed fibrils (PFFs) are to be studied, the protocol used for their preparation is crucial. Conditions used to prepare such samples vary (*Table 1*). It has been previously shown that α-Syn polymorphism may arise when different fibrillization methods are used (*Lv et al., 2012*; *Verasdonck et al., 2016*). NMR has shown to be able to distinguish fibrils in polymorphic mixtures in samples, when the monomer fold differs substantially (*Bousset et al., 2013*; *Gath et al., 2012*; *Gath et al., 2014a*; *Gath et al., 2014b*; *Verasdonck et al., 2016*). Still, NMR could for instance not distinguish between the two assembly polymorphs 2a and 2b. The here reported fibril polymorphism raises questions regarding the structural consistency of recombinant fibrils generated to study α-syn in vitro and calls to investigate the structures resulting from different fibril preparation conditions, or to screen fibril polymorphism in samples prepared by a single preparation protocol, in order to compile a library of α-Syn polymorphs to inform studies using PFFs on cell culture or animal models.

Recently, fibrillar aggregates of Tau protein were purified from human postmortem brain from Alzheimer's, Pick's disease, and chronic traumatic encephalopathy patients, showing variations in Tau fibril conformations between diseases (*Falcon et al., 2018*; *Falcon et al., 2019*;

*Fitzpatrick et al., 2017*), and these fibrils all differed from the in-vitro generated, heparin-induced tau fibrils (*Zhang et al., 2019*). However, to our knowledge, purification of α-Syn fibrils from human brain of Parkinson's disease patients is more challenging. Existing protocols so far have only been able to produce filamentous material that co-fractionated with numerous contaminants (*e.g.*, lipofuscin, amyloid, etc.), including membranes (*Iwatsubo et al., 1996*). Light microscopy and electron microscopy approaches to analyze the ultrastructural composition of LBs show location maps by fluorescence or structural features at the nanometer scale (*Lewis et al., 2019*; *Moors et al., 2018*; *Shahmoradian et al., 2019*). But methods to study the presence and polymorph of α-Syn fibrils in the diseased human brain and cerebrospinal fluid, and the mechanisms by which α-Syn may be causing Parkinson's disease and contribute to progression of the disease, remain to be developed.

## Conclusion

We present here two new structures of α-Syn fibril polymorphs (polymorph 2a (PDB ID 6ssx), and polymorph 2b (PDB ID 6sst)). These differ in their protofilament interfaces but are formed by the same protofibril subunit structure, which is distinct from previously described α-Syn folds.

Our results describe 3D structures very different from previous work and demonstrate how α-Syn amyloid fibrils can reach different cross-ß architectures in spite of having the same amino acid sequence. The structural information from the various α-Syn polymorphs allowed informed hypotheses on how amyloid fibrils may form and how their formation may be related to pathogenicity. More importantly, these structures add to the knowledge of the conformational space of this protein, which is central for structure-based design of imaging tracers or inhibitors of amyloid formation. In this context, a large scale, in vitro study, investigating the structure of α-Syn fibrils produced under different aggregation conditions would prove very informative.

Determination of the structural space of fibril polymorphs, including those of α-Syn carrying disease-relevant mutations, and of α-Syn states purified from diseased human brain, is pivotal to discover whether and how fibrils might form or could be dissolved, and if and how they may interact with affected neurons and contribute to disease.

# Materials and methods

**Key resources table**

| Reagent type (species) or resource | Designation | Source or reference | Identifiers | Additional information |
|---|---|---|---|---|
| Strain, strain background (*E. coli*) | BL21(DE3) | Stratagene | Agilent Technology #200131 | Expression performed in LB medium |
| Transfected construct (pET-3a) | pET3a | Novagen | https://www.addgene.org/vector-database/2637/ | Encoding full length human alpha synuclein asyn (UniProtKB - P37840) with a silent mutagenesis of codon 136 (TAC to TAT) |
| Transfected construct (pRT21) | pRT21 | (*Masuda et al., 2006*) | | Full length human alpha synuclein asyn (UniProtKB - P37840) with a silent mutagenesis of codon 136 (TAC to TAT) |
| Transfected construct (pNatB) | pNatB | (*Johnson et al., 2010*) | http://www.addgene.org/53613/ | Expression of the fission yeast NatB complex - chloramphenicol marker |
| Chemical compound, drug | DEAE sepharose fast flow | GE Healthcare, #17-0709-01 | | |
| Other | Copper/ carbon grids | https://www.quantifoil.com/ | R 2/2 grids | |
| Software, algorithm | UCSF Chimera | (*Pettersen et al., 2004*) | https://www.cgl.ucsf.edu/chimera | RRID:SCR_004097 |

*Continued on next page*

*Continued*

| Reagent type (species) or resource | Designation | Source or reference | Identifiers | Additional information |
|---|---|---|---|---|
| Software, algorithm | SerialEM | (*Mastronarde, 2005*) | https://bio3d.colorado.edu/SerialEM/ | RRID:SCR_017293 |
| Software, algorithm | FOCUS | (*Biyani et al., 2017*) | http://focus-em.org | |
| Software, algorithm | MotionCor2 | (*Zheng et al., 2017*) | https://emcore.ucsf.edu/ucsf-motioncor2 | RRID:SCR_016499 |
| Software, algorithm | RELION 2, 3 | (*Scheres, 2012*; *Zivanov et al., 2018*) | http://www2.mrc-lmb.cam.ac.uk/relion | RRID:SCR_016274 |
| Software, algorithm | COOT | (*Emsley and Cowtan, 2004*) | https://www2.mrc-lmb.cam.ac.uk/personal/pemsley/coot/ | RRID:SCR_014222 |
| Software, algorithm | Molprobity | (*Williams et al., 2018*) | http://molprobity.biochem.duke.edu | RRID:SCR_014226 |

## α-Syn expression and purification

The fibrillary polymorph of WT full length unmodified α-Syn polymorph 2a and 2b were assembled from monomeric α-Syn expressed and purified as described in *Bousset et al. (2013)* and *Gath et al. (2014a)*. Briefly recombinant, wild-type α-Syn was expressed in *E. coli* strain BL21(DE3), transformed with the expression vector pET3a (Novagen) encoding wild-type, full-length α-Syn. Expression was induced by 0.5 mM IPTG for 2 hr, when the bacteria grown in LB medium at 37°C had reached an optical density of 1.0 at 660 nm. Soluble, monomeric α-Syn was purified from the bacterial lysate as previously described (*Ghee et al., 2005*). α-Syn concentration was determined spectrophotometrically using an extinction coefficient of 5960 $M^{-1}*cm^{-1}$ at 280 nm. Pure α-Syn (0.7 mM) in 50 mM Tris-HCl, pH 7.5, 150 mM KCl was filtered through sterile 0.22 µm filters and stored at −80°C.

For preparation of fibrils carrying post-translationally modifications, full-length α-Syn was expressed in competent *Escherichia coli* BL21(DE3) (Stratagene, La Jolla, CA, USA) from the pRT21 expression vector. To acetylate the N-terminus, cells were pre-transfected by pNatB vector coding for the N-terminal acetylase complex (plasmid kindly provided by Daniel Mulvihill, School of Biosciences, University of Kent, Canterbury, UK) (*Johnson et al., 2010*). The various α-Syn forms were purified by periplasmic lysis, ion exchange chromatography, ammonium sulfate precipitation, and gel filtration chromatography as previously described (*Guerrero-Ferreira et al., 2018*; *Huang et al., 2005*; *Luk et al., 2009*). Purified α-Syn was phosphorylated using polo like kinase 2 (PLK2) expressed in *E. coli* BL21-DE3-pLysS, and isolated via its His-tag. Phosphorylated from non-phosphorylated α-Syn was then separated using standard ion exchange and gel filtration chromatography. N-terminally acetylated and phosphorylated α-Syn strains were cleared from endotoxins using one run of Detoxi-Gel Endotoxin Removing Gel (Thermo Scientific) or until endotoxins were below detection level. Protein sequences were verified by tryptic digestion and MALDI mass spectrometry (MS). Alternatively, HPLC/ESI tandem MS was performed to determine total mass. Coomassie blue or silver staining of the SDS PAGE gel and analytical ultracentrifugation were used to determine purity and monodispersity. Protein concentration was measured using the bicinchoninic acid (BCA) assay (Thermo Scientific) with bovine serum albumin as a standard. Purified α-Syn was dialyzed in a 2 kDa Slide-A-Lyzer unit (Thermo Scientific, for max. 3 ml) against HPLC-water (VWR). Aliquots (500 µg) were dispensed into 1.5 ml tubes, frozen on dry ice, and lyophilized for 2 hr in an Eppendorf concentrator (Eppendorf) and stored at −80°C until use.

Disease-linked E46K mutant α-Syn was expressed and purified using a periplasmic purification protocol as described earlier (*Campioni et al., 2014*; *Huang et al., 2005*). Briefly, plasmid pRK172 was co-expressed with N-terminal acetyltransferase B (NatB) complex as described earlier (*Johnson et al., 2010*). Colonies containing both plasmids (NatB and pRK172) were selected using two different antibiotics and grown in 1 liter of lysogeny broth at 37°C. After reaching an optical density of 1.0 at 600 nm, expression was induced with 1 mM IPTG for 5 hr. Cells were harvested and α-Syn collected from the periplasmic space of the cells, using osmotic shock methods described earlier (*Huang et al., 2005*). Protein was further purified using ion exchange chromatography and hydrophobic interaction chromatography (*Campioni et al., 2014*).

## Fibrillization

Full-length, wildtype unmodified α-Syn was incubated at 37°C for one week under continuous shaking in an Eppendorf Thermomixer set at 600 r.p.m., to assemble into fibrillar form. 700 μM α-Syn was assembled in 50 mM Tris-HCl, pH 7.5, 150 mM KCl buffer (*Table 1*).

To prepare fibrils of full-length, phosphorylated, N-terminally acetylated, N-terminally acetylated and the E46K mutant α-Syn, recombinant protein (dialyzed and lyophilized) was diluted to 5 mg/mL in 200 μL of Dulbecco's phosphate buffered saline (DPBS) buffer (Gibco; 2.66 mM KCL, 1.47 mM $KH_2PO_4$, 137.93 mM NaCl, 8.06 mM $Na_2HPO_4$-$7H_2O$ pH 7.0–7.3). After 5 days of incubation at 37°C with constant agitation (1,000 rpm) in an orbital mixer (Eppendorf), reactions were sonicated for 5 min in a Branson 2510 water bath, aliquoted, and stored at −80°C. All fibrils were created in the presence of an air-water interface. The presence of amyloid fibrils was confirmed by thioflavin T fluorimetry and high molecular weight assemblies were visualized by gel electrophoresis.

## Electron microscopy

Cryo-EM grids were prepared using a Leica EM GP automatic plunge freezer (Leica Microsystems) with 80% humidity at 20°C. 3 μL aliquots were applied onto 60 s glow-discharged, 300 mesh, copper Quantifoil grids (R2/1). After blotting, grids were plunge frozen in liquid ethane cooled by liquid nitrogen.

Micrographs were acquired on a Titan Krios (ThermoFisher Scientific) transmission electron microscope, operated at 300 kV and equipped with a Gatan Quantum-LS imaging energy filter (GIF, 20 eV zero loss energy window; Gatan Inc). Images were recorded on a K2 Summit electron counting direct detection camera (Gatan Inc) in dose fractionation mode (50 frames) using the Serial EM software (*Mastronarde, 2005*) at pixel sizes of 0.831 Å or 0.629 Å, and a total dose of ~69 electrons per square Angstrom ($e^-/Å^2$) for each micrograph. Micrographs were processed and analyzed during data collection with FOCUS (*Biyani et al., 2017*), applying drift-correction and dose-weighting using MotionCor2 (*Zheng et al., 2017*). Specific data collection parameters for the various datasets are detailed in *Table 2*.

## Image processing

Computer image processing and helical reconstruction was carried out with RELION 2.1 (*Scheres, 2012*) and RELION 3.0 ß (*Zivanov et al., 2018*), using the methods described in *Zheng et al. (2017)*. Filament selection per micrograph was done manually in RELION 2.1. Segments were extracted with a box size of 280 pixels and an inter-box distance of 28 pixels. A summary of the number of micrographs and segments that went into the various steps of processing are presented in *Table 2*. After 2D classification with a regularization value of T = 2, 2D class averages with a visible separation of individual rungs were selected for further processing. For class averages formed by segments from polymorph 2, the calculated power spectra showed a meridional peak intensity (Bessel order n = 0) at the layer line of 1/ (4.8 Å). For polymorph 2b, power spectra showed peak intensities on both sides of the meridian (Bessel order n = 1). This is the result of an approximate $2_1$ screw symmetry between α-Syn subunits on the two protofilaments (*Figure 1—figure supplement 1*). The best 2D classes were selected for a round of 3D classification with T = 8 and optimization of the helical twist and rise using a *helical_z_percentage* parameter (*He and Scheres, 2017*) of 10%. A cylinder generated via the helix toolbox tool was used as initial model. This resulted in three classes where ß-sheets perpendicular to the fibril axis were clearly separated. Segments contributing to classes 1 and 2, which corresponded to α-Syn polymorphs 2a and 2b, respectively, were then processed separately. The 3D maps from their respective classes were used as initial models after applying a low-pass filter to 30 Å. Then, a 3D classification with a single class (K = 1) and T = 20, which has allowed the successful reconstruction of amyloid filaments, was carried out (*Falcon et al., 2018*; *Fitzpatrick et al., 2017*; *Guerrero-Ferreira et al., 2018*).

The 3D auto-refine procedure in RELION 3.0, with optimization of helical twist and rise, resulted in structures with overall resolutions of 3.34 Å (α-Syn polymorph 2a) and 3.75 Å (α-Syn polymorph 2b). Post-processing with soft-edge masks and estimated map sharpening *B*-factors of −67.1 and −76.4 $Å^2$, respectively, gave maps with resolutions of 3.0 Å (α-Syn polymorph 2a) and 3.4 Å (α-Syn polymorph 2b) (by the FSC 0.143 criterion). Local resolution values and local-resolution-filtered maps were obtained in RELION 3.0. (*Table 2*, *Figure 1—figure supplement 1*).

## Model building and refinement

A model for the α-Syn polymorph 2a fibril was built into the RELION local resolution-filtered map with COOT (*Emsley and Cowtan, 2004*), by conserving as many secondary structure elements as possible from our previous α-Syn polymorph one model (PDB ID 6h6b) (*Guerrero-Ferreira et al., 2018*), together with the use of secondary structure information derived from ssNMRs, which served as an initial model. The structure was then refined against the same map with PHENIX real space refine (*Afonine et al., 2013*) using rotamer and Ramachandran restraints, and non-crystallographic symmetry and beta strand geometry constraints. The building and refinement of the α-Syn polymorph 2b model was more challenging due to the lower resolution of the map. An initial structure was built into the α-Syn polymorph 2b local resolution-filtered map by fitting of the α-Syn polymorph 2a model using COOT. To successfully refine the structure, it was necessary to generate secondary structure restraints in Phenix, based on CA backbone and intermolecular beta sheets to refine the structure. All structures were validated using Molprobity (*Williams et al., 2018*). Figures were prepared using UCSF Chimera (*Pettersen et al., 2004*).

## NMR spectroscopy

NMR spectra were recorded at 20.0 T static magnetic field using 3.2 mm rotors and a triple-resonance probe. The reproducibility of the sample preparation was previously verified with 20 ms DARR fingerprint spectra (*Gath et al., 2014a*). The secondary chemical-shift analysis was based on the sequential assignments (BMRB accession code 18860) and was presented before (*Gath et al., 2014a*; *Gath et al., 2014b*). The PAR spectrum (*De Paëpe et al., 2008*; *Lewandowski et al., 2007*) was recorded using a mixing time of 8 ms. The assignments for the S87/A17-19 cross peaks are unambiguous within a range of 0.15 ppm, corresponding to about ½ of the $^{13}$C line width. Assignments of restraints given for reference on the full aliphatic region of the PAR spectrum in the *Figure 1—figure supplement 3* are mostly ambiguous, which prevented structure determination by NMR. Ambiguities were lifted by comparison to the here determined cryo-EM structure.

## Data availability

Raw cryo-EM micrographs are available in EMPIAR, entry numbers EMPIAR-10323. The 3D maps are available in the EMDB, entry numbers EMD-10307 (α-Syn polymorph 2a) and EMD-10305 (α-Syn polymorph 2b). Atomic coordinates are available at the PDB with entry numbers PDB 6SSX (α-Syn polymorph 2a) and PDB 6SST (α-Syn polymorph 2b).

## Acknowledgements

We thank Liz Spycher, Jana Ebner, Alexandra Kronenberger, Daniel Schlatter, Daniela Huegin, Ralph Thoma, Christian Miscenic, Martin Siegrist, Sylwia Huber, Arne Rufer, Eric Kusznir, Peter Jakob, Tom Dunkley, Joerg Hoernschmeyer, and Johannes Erny at Roche for their technical support to clone, express, purify and characterize the different forms of α-Syn; Kenneth N Goldie, Lubomir Kovacik and Ariane Fecteau-Lefebvre for support in cryo-EM. Calculations were performed using the high-performance computing (HPC) infrastructure administered by the scientific computing center at the University of Basel (sciCORE; http://scicore.unibas.ch). The Novo Nordisk Foundation Center for Protein Research is supported financially by the Novo Nordisk Foundation (NNF14CC0001). NMIT is a member of the Integrative Structural Biology Cluster (ISBUC) at the University of Copenhagen. This work was in part supported by the Synapsis Foundation Switzerland, the Heidi-Seiler Stiftung Foundation, and the Swiss National Science Foundation (grants CRSII3_154461, CRSII5_177195. 20020_178792 and NCCR TransCure), and the French Agence Nationale de la Recherche Scientifique (ANR-12-BS08-0013-01. ANR-11-LABX-0048 and ANR-11-IDEX-0007). AAA, LB and RM received funding from the European Union's Horizon 2020 research and innovation programme under grant agreement No. 116060 (IMPRiND), the Swiss State Secretariat for Education, Research and Innovation (SERI) under contract number 17.00038, the Fondation Bettencourt Schueller, The Fondation pour la Recherche Médicale (Contract DEQ 20160334896),The Fondation Simone et Cino Del Duca of the Institut de France and the EC Joint Program on Neurodegenerative Diseases (TransPathND, ANR-17-JPCD-0002–02 and Protest-70, ANR-17-JPCD-0005–01), by the French Infrastructure for

Integrated Structural Biology (FRISBI) [ANR-10-INSB-05–01]. The opinions expressed and arguments employed herein do not necessarily reflect the official views of these funding bodies.

## Additional information

### Competing interests

Daniel Mona, Markus Britschgi, Matthias E Lauer: employee at Roche and may additionally hold Roche stock/stock options. The other authors declare that no competing interests exist.

### Funding

| Funder | Grant reference number | Author |
| --- | --- | --- |
| Novo Nordisk Foundation | NNF14CC0001 | Nicholas MI Taylor |
| Synapsis Foundation – Alzheimer Research Switzerland | | Ricardo Guerrero-Ferreira Henning Stahlberg |
| Heidi-Seiler Stiftung | | Ricardo Guerrero-Ferreira Henning Stahlberg |
| Swiss National Science Foundation | CRSII3_154461 | Ricardo Guerrero-Ferreira Nicholas MI Taylor Henning Stahlberg |
| Swiss National Science Foundation | CRSII5_177195 | Ricardo Guerrero-Ferreira Henning Stahlberg |
| Swiss National Science Foundation | 20020_178792 | Beat H Meier |
| Agence Nationale de la Recherche | ANR-12-BS08-0013-01 | Ronald Melki Luc Bousset |
| Labex | ANR-11-LABX-0048 | Anja Böckmann |
| EU Framework Programme for Research and Innovation H2020 | 116060 IMPRiND | Ana-Andreea Arteni Ronald Melki Luc Bousset |
| SERI | 17.00038 | Ana-Andreea Arteni Luc Bousset Ronald Melki |
| Fondation Bettencourt Schueller | | Ana-Andreea Arteni Luc Bousset Ronald Melki |
| Fondation pour la Recherche Médicale | Contract DEQ 20160334896 | Ana-Andreea Arteni Luc Bousset Ronald Melki |
| Fondation Simone et Cino Del Duca | TransPathND ANR-17-JPCD-0002-02 | Luc Bousset Ronald Melki |
| EU Joint Programme – Neurodegenerative Disease Research | Protest-70 ANR-17-JPCD-0005-01 | Ana-Andreea Arteni Luc Bousset Ronald Melki |
| French Infrastructure for Integrated Structural Biology | ANR-10-INSB-05-01 | Ana-Andreea Arteni Luc Bousset Ronald Melki |
| University of Lyon | Investissements d'Avenir ANR-11-IDEX-0007 | Anja Böckmann |
| Swiss National Science Foundation | NCCR TransCure | Nicholas MI Taylor Ricardo Guerrero-Ferreira Henning Stahlberg |

The funders had no role in study design, data collection and interpretation, or the decision to submit the work for publication.

## Author contributions

Ricardo Guerrero-Ferreira, Conceptualization, Data curation, Formal analysis, Validation, Investigation, Visualization, Methodology, Writing - original draft, Writing - review and editing; Nicholas MI Taylor, Data curation, Formal analysis, Validation, Investigation, Visualization, Writing - review and editing, Structural calculation, building of the structural model; Ana-Andreea Arteni, Validation, Investigation, Screen the assemblies for optimization, optimize cryo-EM freezing conditions and set up experimental conditions for particle imaging, Acquire the first high resolution Cryo-EM images of the "fibril" type of assemblies; Pratibha Kumari, Validation, Investigation, Data interpretation and discussion; Daniel Mona, Matthias E Lauer, Investigation, Methodology, Expressed, purified and analyzed recombinant human alpha-synuclein (wt, N-acetylated, pSer129 aSyn, C-terminally truncated forms) and performed initial biochemical and biophysical characterization and quality control of generated fibrils; Philippe Ringler, Data curation, Formal analysis, Investigation, Data interpretation and discussion; Markus Britschgi, Resources, Data curation, Investigation, Methodology, Expressed, purified and analyzed recombinant human alpha-synuclein (wt, N-acetylated, pSer129 aSyn, C-terminally truncated forms) and performed initial biochemical and biophysical characterization and quality control of generated fibrils; Ali Makky, Formal analysis, Investigation, Methodology, Acquire and interpret AFM data on alpha-synuclein; Joeri Verasdonck, Data curation, Formal analysis, Investigation, Acquire and interpret Solid state NMR data; Roland Riek, Investigation, Methodology, Data interpretation and discussion; Ronald Melki, Resources, Formal analysis, Supervision, Funding acquisition, Investigation, Visualization, Methodology, Writing - review and editing; Beat H Meier, Conceptualization, Resources, Supervision, Investigation, Methodology, Writing - review and editing; Anja Böckmann, Conceptualization, Supervision, Investigation, Writing - review and editing, Solid-state NMR studies, Supervision, Building of the structural model; Luc Bousset, Conceptualization, Resources, Formal analysis, Validation, Investigation, Visualization, Methodology, Writing - original draft, Writing - review and editing, Expressed, purified and analyzed recombinant human alpha-synuclein (wt, unmodified form) generated fibrils; Henning Stahlberg, Conceptualization, Resources, Software, Supervision, Funding acquisition, Validation, Investigation, Methodology, Writing - original draft, Project administration, Writing - review and editing

## Author ORCIDs

Ricardo Guerrero-Ferreira https://orcid.org/0000-0002-3664-8277
Nicholas MI Taylor http://orcid.org/0000-0003-0761-4921
Ana-Andreea Arteni https://orcid.org/0000-0001-6462-905X
Philippe Ringler https://orcid.org/0000-0003-4346-5089
Markus Britschgi https://orcid.org/0000-0001-6151-4257
Matthias E Lauer https://orcid.org/0000-0003-3252-8718
Beat H Meier http://orcid.org/0000-0002-9107-4464
Anja Böckmann https://orcid.org/0000-0001-8149-7941
Henning Stahlberg https://orcid.org/0000-0002-1185-4592

## Decision letter and Author response

Decision letter https://doi.org/10.7554/eLife.48907.sa1
Author response https://doi.org/10.7554/eLife.48907.sa2

---

# Additional files

## Supplementary files

• Transparent reporting form

## Data availability

Raw cryo-EM micrographs are available in EMPIAR, entry numbers EMPIAR-10323. The 3D maps are available in the EMDB, entry numbers EMD-10307 (α-Syn polymorph 2a) and EMD-10305 (α-Syn-polymorph 2b). Atomic coordinates are available at the PDB with entry numbers PDB 6SSX (α-Syn polymorph 2a) and PDB 6SST (α-Syn polymorph 2b).

The following datasets were generated:

| Author(s) | Year | Dataset title | Dataset URL | Database and Identifier |
|---|---|---|---|---|
| Ricardo Guerrero-Ferreira, Nicholas MI Taylor, Ana-Andreea Arteni, Pratibha Kumari, Daniel Mona, Philippe Ringler, Markus Britschgi, Matthias E Lauer, Ali Makky, Joeri Verasdonck, Roland Riek, Ronald Melki, Beat H Meier, Anja Böckmann, Luc Bousset, Henning Stahlberg | 2019 | aSyn polymorph 2a | https://www.ebi.ac.uk/pdbe/entry/emdb/EMD-10307 | Electron Microscopy Data Bank, EMD-10307 |
| Ricardo Guerrero-Ferreira, Nicholas MI Taylor, Ana-Andreea Arteni, Pratibha Kumari, Daniel Mona, Philippe Ringler, Markus Britschgi, Matthias E Lauer, Ali Makky, Joeri Verasdonck, Roland Riek, Ronald Melki, Beat H Meier, Anja Böckmann, Luc Bousset, Henning Stahlberg | 2019 | aSyn polymorph 2a | https://pdbe.org/6ssx | Electron Microscopy Data Bank, 6SSX |
| Ricardo Guerrero-Ferreira, Nicholas MI Taylor, Ana-Andreea Arteni, Pratibha Kumari, Daniel Mona, Philippe Ringler, Markus Britschgi, Matthias E Lauer, Ali Makky, Joeri Verasdonck, Roland Riek, Ronald Melki, Beat H Meier, Anja Böckmann, Luc Bousset, Henning Stahlberg | 2019 | aSyn polymorph 2b | https://www.ebi.ac.uk/pdbe/entry/emdb/EMD-10305 | Electron Microscopy Data Bank, EMD-10305 |
| Ricardo Guerrero-Ferreira, Nicholas MI Taylor, Ana-Andreea Arteni, Pratibha Kumari, Daniel Mona, Philippe Ringler, Markus Britschgi, Matthias E Lauer, Ali Makky, Joeri Verasdonck, Roland Riek, Ronald Melki, Beat H Meier, Anja Böckmann, Luc Bousset, Henning Stahlberg | 2019 | aSyn polymorph 2b | https://pdbe.org/6sst | Electron Microscopy Data Bank, 6SST |
| Ricardo Guerrero-Ferreira, Nicholas MI Taylor, Ana-Andreea Arteni, Pratibha Kumari, Daniel Mona, Philippe Ringler, Markus Britschgi, Matthias E Lauer, Ali Makky, | 2019 | aSyn polymorph 2a and 2b electron microscopy images | https://www.ebi.ac.uk/pdbe/emdb/empiar/entry/10323 | Electron Microscopy Data Bank, EMPIAR-10323 |

Joeri Verasdonck,
Roland Riek, Ronald
Melki, Beat H Meier,
Anja Böckmann, Luc
Bousset, Henning
Stahlberg

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
