## [Decision Letter]

**Acceptance summary:**

This paper describes two new polymorphs of in vitro assembled α-synuclein filaments. The new polymorphs termed 2a and 2b were generated under different growth conditions in comparison to previously determined polymorphs 1. Most surprisingly, the polymorphs of group 2 fold into bent β-arches conformation of similar outline of polypeptide path but involving different segments of the structure combined into the β-arch fold. Although the two polymorphs of group 2 share the same core structure, they differ in their relative arrangement. Although the disease-relevance of the new structures is unclear, these results help exploring the conformational variability in α-synuclein amyloid filaments, and the discussion on how different in-vitro aggregation conditions might lead to different polymorphs was considered useful.

**Decision letter after peer review:**

Thank you for submitting your article "Two new polymorphic structures of α-synuclein solved by cryo-electron microscopy" for consideration by *eLife*. Your article has been reviewed by three peer reviewers, including Sjors HW Scheres as the Reviewing Editor and Reviewer #1, and the evaluation has been overseen by John Kuriyan as the Senior Editor.

The reviewers have discussed the reviews with one another and the Reviewing Editor has drafted this decision to help you prepare a revised submission.

The reviewers had an extended discussion whether this manuscript is suitable for publication in *eLife*. On the one hand, the disease-relevance of the new structures is unclear; on the other hand, these results help exploring the conformational variability in α-synuclein amyloid filaments. However, it was felt that potentially the most significant contribution of this manuscript is the discussion on how different in-vitro aggregation conditions might lead to different polymorphs. The paper is well written, but the points below need to be addressed before publication.

Essential revisions:

1) It would be good if the authors could extend on their rationale how different in-vitro aggregation conditions might lead to different polymorphs. For example, could they make predictions whether new, unseen conditions would lead to polymorphs of type 1 or 2? For example, by testing different (poly)anions? At the very least, (the feasibility of) such ideas should be discussed. However, if such predictions could even be tested experimentally, then this would further strengthen the paper. (This would not necessarily require cryo-EM reconstructions; possibly (2D class averages of) negative stain images would suffice.)

2) The authors mention two-fold and approximate 2_1 screw symmetry for the two polymorphs. However, these symmetries do not seem to have been applied to the maps. This should be rectified. Also, there seem to be main-chain breaks between Ala69 and Val70 in both structures. This cannot be due to conformational heterogeneity, as the entire neighbourhood of these residues is very well ordered. Therefore, these are probably the results of local minima in the helical refinement. This needs to be resolved; hopefully applying the correct symmetry will help. It would help the reviewing process if the authors could submit unfiltered and unmasked half-maps with their revision. Also, hopefully in the new maps the observation that C-terminal density is interpreted with side chains (90-95) when no side chain is clearly visible will be no longer relevant. If it does remain relevant, side chains should be removed from the model.

3) The 2a map contains more detail to justify accurate model building than the 2b map. Therefore, the authors should detail their strategy of model building for map 2b more clearly if their model building was primarily based on map 2a. In this context, it is not clear how the described differences are due to the low resolution or represent true structural differences. In the two polymorphs, the authors claim that they have an identical fold. Inspecting the atomic models, it is clear that the secondary structure assignments of β-sheets from the two polymorphs are not identical. Based on the Materials and methods statement: "To successfully refine the structure, it was necessary to generate secondary structure restraints to refine the structure using.", it would be illuminating to detail the restraints more explicitly and why the refinement of model 2b resulted in a more relaxed conformation with less β-sheet content. Possibly, it could be more appropriate to restrain them more strongly in model 2b due to lower resolution. In addition, the different chains within a PDB file are not identical. During the refinement, all chains need to be symmetry restrained to yield the same conformation. This could have been a consequence of the non-imposed symmetry on the map, but it should be resolved.

4) A rise of nearly 5A seems too high for energetically stable β-sheet formation. Is the pixel size correct? The authors should provide convincing evidence for absolute pixel calibration, or in the absence of such evidence, re-scale the pixel to the expected rise of 4.7-8A.

5) Are the authors convinced the handedness of their structure is correct? What was it based on?

6) Given the disease relevance of the results is uncertain, it would be good to focus the Introduction more on synucleinopathies in general, as opposed to the current focus on Parkinson's disease (PD). Since this study elaborates on the structural polymorphism of assembled α-synuclein, the clinical heterogeneity of synucleinopathies should be emphasised more. In addition, at least human brain-derived filaments from PD should be mentioned and their morphological similarities/differences to the in vitro assembled polymorphs discovered to date should be explained (e.g. R.A. Crowther et al., 2000).

7) Figure 3 contains too low resolution maps to convincingly make the point that these structures adopt the same main chain trace as the polymorph 2a. Especially Figure 3B and 3C look very artefactual. Perhaps X-Y slices could be used, otherwise these reconstructions should be removed from the paper. If the authors are keen to keep these results in, the reconstructed maps should be submitted alongside the revision so that a better informed decision can be made.

---

## [Author Response]

Essential revisions:1) It would be good if the authors could extend on their rationale how different in-vitro aggregation conditions might lead to different polymorphs. For example, could they make predictions whether new, unseen conditions would lead to polymorphs of type 1 or 2? For example, by testing different (poly)anions? At the very least, (the feasibility of) such ideas should be discussed. However, if such predictions could even be tested experimentally, then this would further strengthen the paper. (This would not necessarily require cryo-EM reconstructions; possibly (2D class averages of) negative stain images would suffice.)

The use of polyanions has been widely described by Fink and Uversky for over two decades. They showed that assembly conditions and additives lead to different kinetics and morphologies. However, their work with poly-anions was not intended to obtain non-polymorphic samples suitable for structural studies (cryo EM, NMR, X-ray). Instead, we screened for α synuclein assembly conditions that would produce a single fold of a-syn fibrils, suitable for structure-function investigations.

Indeed, we did identify assembly conditions of WT human α synuclein leading to distinct and yet undetermined α synuclein amyloid folds (Bousset et al., 2013; Makky et al., 2016; Verasdonck et al., 2016). In this work, the ribbon polymorph displayed a radically different distribution of the β strands along the synuclein sequence and lacked a defined helical repeat, not seen in any published structure of α synuclein. Polymorph fibril 91 showed at least a new quaternary fold with a helical assembly of protofilaments with an apparent width of more than 50nm.

Trying to provide a rationale for amyloid folding would be, in our view, even more challenging than trying to explain protein folding of globular protein. The interplay between intramolecular folding on one side, and templating by intermolecular interactions on the other side, is still poorly understood. We prefer not to speculate on mechanisms governing amyloid folding. With that said, a large scale project to investigate the formation of α-synuclein fibrils under different in-vitro aggregation conditions is indeed an interesting idea, but one that would require time and effort, which is outside the scope of our study.

2) The authors mention two-fold and approximate 2_1 screw symmetry for the two polymorphs. However, these symmetries do not seem to have been applied to the maps. This should be rectified. Also, there seem to be main-chain breaks between Ala69 and Val70 in both structures. This cannot be due to conformational heterogeneity, as the entire neighbourhood of these residues is very well ordered. Therefore, these are probably the results of local minima in the helical refinement. This needs to be resolved; hopefully applying the correct symmetry will help. It would help the reviewing process if the authors could submit unfiltered and unmasked half-maps with their revision. Also, hopefully in the new maps the observation that C-terminal density is interpreted with side chains (90-95) when no side chain is clearly visible will be no longer relevant. If it does remain relevant, side chains should be removed from the model.

We thank the reviewers for these constructive comments. We have re-processed the maps, applying the approximate 2_1 screw symmetry. Polymorph 2a was solved with a helical rise of 4.8Å and a helical twist of -0.8°. Polymorph 2b has a helical rise of 2.4Å and a helical twist of 179.6°. After reprocessing the data with the correct symmetry applied, the main-chain break between Ala69 and Val70 was resolved. Both unfiltered and unmasked half-maps are submitted with this revision.

3) The 2a map contains more detail to justify accurate model building than the 2b map. Therefore, the authors should detail their strategy of model building for map 2b more clearly if their model building was primarily based on map 2a. In this context, it is not clear how the described differences are due to the low resolution or represent true structural differences. In the two polymorphs, the authors claim that they have an identical fold. Inspecting the atomic models, it is clear that the secondary structure assignments of β-sheets from the two polymorphs are not identical. Based on the Materials and methods statement: "To successfully refine the structure, it was necessary to generate secondary structure restraints to refine the structure using.", it would be illuminating to detail the restraints more explicitly and why the refinement of model 2b resulted in a more relaxed conformation with less β-sheet content. Possibly, it could be more appropriate to restrain them more strongly in model 2b due to lower resolution. In addition, the different chains within a PDB file are not identical. During the refinement, all chains need to be symmetry restrained to yield the same conformation. This could have been a consequence of the non-imposed symmetry on the map, but it should be resolved.

The lower resolution of the map for polymorph 2b impacts the quality of the model, the missing interaction between the two protofilaments around residues 57 also reveals the flexibility of this loop. The model for polymorph 2b was built by starting with the polymorph 2a model, and inspecting the backbone in the 2b density map. The lower resolution of the 2b map leads to weaker restrains during refinement, therefore we imposed intermolecular hydrogen bonds during the refinement (using the “SHEET atom 1 --- Atom 2” command in the PDB header, with visually identified H-bonds, Phenix refinement procedure allocate restrain for all H bonds that are closer than 2.9A apart).

The 10 chains within the PDB were not restrained due to the fact that the map is not of the same quality along fibril axis, and the two protofilaments were not restrained either. The reason for this is that the masking procedure leads to maps of lower quality at the edges of the mask, and that we need multiple layers to refine the intermolecular hydrogen bonds that run parallel to the fibril axis.

We can impose a symmetry or dock polymorph 2a structure, but this would erase minor differences that we can detect if symmetry averaging or restrain are not imposed.

4) A rise of nearly 5A seems too high for energetically stable β-sheet formation. Is the pixel size correct? The authors should provide convincing evidence for absolute pixel calibration, or in the absence of such evidence, re-scale the pixel to the expected rise of 4.7-8A.

We thank the reviewers for this recommendation. We have verified and corrected the pixel size to calibrated values. It now is 0.629 Å. The corrected maps now have a rise of 4.8Å for polymorph 2a and 2.4Å for polymorph 2b.

5) Are the authors convinced the handedness of their structure is correct? What was it based on?

The handedness was analyzed by AFM imaging, as also done previously (Guerrero-Ferreira et al., 2018).

6) Given the disease relevance of the results is uncertain, it would be good to focus the Introduction more on synucleinopathies in general, as opposed to the current focus on Parkinson's disease (PD). Since this study elaborates on the structural polymorphism of assembled α-synuclein, the clinical heterogeneity of synucleinopathies should be emphasised more. In addition, at least human brain-derived filaments from PD should be mentioned and their morphological similarities/differences to the in vitro assembled polymorphs discovered to date should be explained (e.g. R.A. Crowther et al., 2000).

Thank you for this suggestion. We have modified the Introduction accordingly to recognize previous morphological characterization of α-synuclein fibrils.

7) Figure 3 contains too low resolution maps to convincingly make the point that these structures adopt the same main chain trace as the polymorph 2a. Especially Figure 3B and 3C look very artefactual. Perhaps X-Y slices could be used, otherwise these reconstructions should be removed from the paper. If the authors are keen to keep these results in, the reconstructed maps should be submitted alongside the revision so that a better informed decision can be made.

We thank the reviewers for these comments, and have replaced Figure 3 with 2d projections of the filament cross-sections.